# Could Horizontal Gene Transfer Explain 5S rDNA Similarities Between Frogs and Worm Parasites?

**DOI:** 10.3390/biom15071001

**Published:** 2025-07-12

**Authors:** Kaleb Pretto Gatto, Cintia Pelegrineti Targueta, Stenio Eder Vittorazzi, Luciana Bolsoni Lourenço

**Affiliations:** 1Laboratório de Estudos Cromossômicos, Departamento de Biologia Estrutural e Funcional, Instituto de Biologia, Universidade Estadual de Campinas, Campinas 13083-863, São Paulo, Brazil; kaleb.gatto@ufpr.br; 2Laboratório de Citogenética Evolutiva e Conservação Animal, Departamento de Genética, Setor de Ciências Biológicas, Universidade Federal do Paraná, Curitiba 80230-901, Paraná, Brazil; 3Laboratório de Genética e Biodiversidade, Departamento de Genética, Instituto de Ciências Biológicas, Universidade Federal de Goiás, Goiânia 74690-900, Goiás, Brazil; cincintia@ufg.br; 4Departamento de Biologia, Universidade do Estado do Mato Grosso, Tangará da Serra 78690-000, Mato Grosso, Brazil; stenio.vittorazzi@gmail.com

**Keywords:** Anura, rDNA, lateral gene transfer, Nematoda, Platyhelminthes

## Abstract

Horizontal gene transfer (HGT), the non-Mendelian transfer of genetic material between organisms, is relatively frequent in prokaryotes, whereas its extent among eukaryotes remains unclear. Here, we raise the hypothesis of a possible cross-phylum HGT event involving 5S ribosomal DNA (rDNA). A specific type of 5S rDNA sequence from the anuran *Xenopus laevis* was highly similar to a 5S rDNA sequence of the genome of its flatworm parasite *Protopolystoma xenopodis*. A maximum likelihood analysis revealed phylogenetic incongruence between the gene tree and the species trees, as the 5S rDNA sequence from *Pr. xenopodis* was grouped along with the sequences from the anurans. Sequence divergence analyses of the gene region and non-transcribed spacer also agree with an HGT event from *Xenopus* to *Pr. xenopodis*. Additionally, we examined whether contamination of the *Pr. xenopodis* genome assembly with frog DNA could explain our findings but found no evidence to support this hypothesis. These findings highlight the possible contribution of HGT to the high diversity observed in the 5S rDNA family.

## 1. Introduction

Horizontal or lateral gene transfer (HGT/LGT), the nongenealogical exchange of DNA among organisms, may be an important source of genetic variation, introducing genetic novelties into the acceptor genome, such as new genes or new variants of a given gene or repetitive DNA sequence [1,2,3,4,5,6].

Several cases of gene transfer from prokaryotes to prokaryotes have been documented, and there is no doubt about the impact of HGT on the evolution of prokaryotes [7,8], with numerous examples of the transfer of antibiotic resistance genes by HGT [9]. Although less frequent than prokaryote-to-prokaryote HGT, prokaryote-to-eukaryote HGT has also been commonly reported and implicated in the origin of several functional genes in eukaryotes [9,10,11,12]. In contrast, HGT from eukaryotes to prokaryotes and, particularly, HGT between eukaryotes are less understood and, sometimes, controversial [13,14,15,16].

Some eukaryotic traits, such as the presence of a nuclear envelope and the sequestration of germline cells in the case of bilaterian animals [17], are relevant obstacles to nuclear gene transfer among eukaryotes and, therefore, may explain the lower frequency of eukaryote–eukaryote HGT. Nevertheless, because HGT is first inferred from incongruities observed between a particular gene/DNA sequence tree and a species tree, the extent of eukaryote-to-eukaryote HGT may be underestimated owing to the scarcity of information about genomes across all eukaryotic phyla. Recent advances in genome sequencing, computational analysis, and analysis of transposable elements have considerably increased the number of HGT reports in recent years across all trees of life [3,18,19]. Therefore, the extent of eukaryote-to-eukaryote HGT and the impact of HGT on eukaryote evolution remain largely unknown.

Among the documented cases of functional nuclear gene transfer within eukaryotes, several reports refer to HGT between fungi [20], but other examples include fungi to insects (the pea aphid *Acyrthosiphon pisum*) [21], fungi to Lepidoptera [22], bryophytes to ferns [23], plants to Nematoda [24], autotrophic to heterotrophic plants [25], and fish to fish [26] HGT. Recent studies on transposable elements [18,27,28,29] have provided increasing evidence of HGT across phyla, including vertebrates. However, to date, there is only limited evidence for nuclear gene transfer across phyla involving chordates. Two examples from chordates involve phylogenetically distantly related species of fish, with one of them involving a lateral transfer of the type II antifreeze protein gene [26,30] and another involving the transfer of 5S ribosomal DNA (rDNA) [31].

Considering that ecological relationships, such as endosymbiosis and parasitism, may favor HGT because of the longstanding and close interaction of cells from the involved organisms [4,32], we searched for evidence of HGT between frogs and parasitic worms, including *Xenopus* species of frogs and their parasitic monogenean *Protopolystoma xenopodis*. We used 5S rDNA sequences in this analysis because (a) there are numerous frog and worm 5S rDNA/rRNA gene sequences deposited in public databases; (b) although the 5S rDNA transcribing region is evolutionarily conserved, substantial divergence of 5S rDNA occurs between vertebrates and other metazoans [33]; and (c) the 5S rDNA non-transcribing spacers (NTSs) are expected to vary substantially between species and even within the genome of a given species [34,35,36,37,38]. Based on our findings, we examine the hypothesis of HGT between frogs and parasitic worms, expanding the number of candidate cases of HGT between eukaryotic phyla and providing new insights into the potential role of HGT in 5S rDNA evolution.

## 2. Materials and Methods

### 2.1. Data Acquisition

We obtained 5S rDNA sequences from several species of flatworms, nematodes, and frogs in GenBank (www.ncbi.nlm.nih.gov/genbank/; accessed from 23 March 2018 to 22 December 2024), WormBase ParaSite (www.parasite.wormbase.org; accessed from 23 March 2018 to 22 December 2024), and/or the 5S rRNA database [39]. All the sequence sources and accession numbers are presented in Appendix A. Additionally, we retrieved sequences of 5S rRNA genes from genome assemblies of *Protopolystoma xenopodis* [40] and used 5S rDNA sequences that were previously retrieved from genome assemblies of anurans species by Targueta et al. [41]. We also analyzed the short-read libraries employed to assemble the *Pr. xenopodis* genome, which were made using individuals in different developmental stages (eggs, larvae and adult) [40].

### 2.2. Phylogenetic and Similarity Analysis of 5S rRNA Genes

We generated a data matrix compiling all the presumed transcribed regions of 5S rDNA (retrieved from 5S rDNA clones and genome assemblies) along with sequences from the annotated 5S rRNA gene. First, we aligned the 5S rDNA and/or 5S rRNA gene of each species or genus (e.g., all *Physalaemus cuvieri* type I 5S rDNA) to determine the different haplotypes using DnaSP v.6.12.01 [42]. Second, the 5S rRNA gene region from each haplotype was compiled into a single matrix, and all the sequences were aligned using the ClustalW algorithm [43] implemented in BioEdit v.7.2.5 [44]. Ambiguously aligned regions were corrected manually in BioEdit. The resulting matrix contained 529 sequences of anurans, flatworms, and nematodes. Third, we used this aligned matrix to conduct a maximum likelihood (ML) analysis under the Kimura-2-parameter model in MEGA X v.10.2 [45] and estimated node support via bootstrap analysis of 1000 pseudoreplicates. We estimated the p-distance between the presumed transcribed region of the 5S rDNA sequences in MEGA X, treating alignment gaps and missing data as pairwise deletions.

### 2.3. Analysis of 5S rDNA Non-Transcribing Spacers in Frogs and Worms

Because we found a sequence identical to the 5S RNA gene of *Protopolystoma xenopodis* in the genome of *Xenopus laevis* (see Section 3), we investigated whether the non-transcribed regions associated with these sequences were also similar to one another. We used the 5S rRNA gene sequence of *X. laevis* (GenBank accession number: J01009) that was 100% identical to one sequence of *Pr. xenopodis* as a query in BLAST searches^86^ against the *Pr. xenopodis* genome assembly (assembly accession number: GCA_900617795). We then used the contig from the *Pr. xenopodis* genome assembly identified in this analysis as a query to search for similar sequences in the genomes of other flatworms (the species are listed in Appendix A). Additionally, we used the *Pr. xenopodis* contigs with 5S rRNA gene annotation in BLAST searches against genome assemblies of *X. laevis* (GCA_001663975.1), *X. tropicalis* (GCA_000004195.3), and *Nanorana parkeri* (GCA_000935625.1).

Since the *Pseudis tocantins* type I 5S rRNA gene [46] showed high similarity to the 5S rRNA genes of *Bursaphelenchus xylophilus* and *Subanguina moxae* (see Section 3), we conducted BLAST searches using the *Ps. tocantins* sequence KX170899 as a query against genome assemblies of *B. xylophilus* species available at the WormBase ParaSite (PRJEA64437 and PRJEB40022). BLAST searches were also performed using only the NTS of type I 5S rDNA of *Ps. tocantins* against the NCBI nucleotide collection and genome assemblies available in WormBase ParaSite.

### 2.4. Assessment of DNA Contamination Hypothesis

In addition to horizontal transfer, another possible explanation for the high similarity found between 5S rDNA sequences from frogs and worms is DNA contamination. If the hypothesis of DNA contamination is correct, we would expect to find evidence for it when analyzing other DNA sequences distinct from 5S rDNA. Considering that one of the 5S rRNA gene sequences found in the genome assembly of *Protopolystoma xenopodis* was very similar to sequences found in distinct studies of *Xenopus laevis* (sequences retrieved from the genome assembly GCA_001663975.1 and sequence J01009, obtained from DNA fragments isolated via CsCl density gradient centrifugation) and in *X. borealis* (V01426), it is plausible that the *Pr. xenopodis* database is contaminated with *Xenopus* DNA. To test whether the *Pr. xenopodis* database provides further evidence of frog contaminant DNA, we performed BLASTn v.2.16.0 searches using eukaryotic-conserved multigene families (U1 snRNA gene, 40S rDNA genes and intergenic spacer, H3 histone gene), conserved or vertebrate-specific single-copy genes (Rag-1 and Rhod), and some representative transposable elements (high-copy-number TEs: Harbinger and Tc1-Mariner; low-copy-number TEs: Gypsy, DIRS, and CR1) from *X. laevis* (Appendix A) as queries against the *Pr. xenopodis* genome assembly. If the highest score hits of some of these sequences showed high values for similarity and query cover and a low e-value, we would consider that *Pr. xenopodis* sequenced libraries could contain *Xenopus* contaminant DNA. We also compared the highest score hits found in these searches with those found using the 5S rDNA as the query.

Additionally, to evaluate whether the *Pr. xenopodis* genome assembly is contaminated with *Xenopus* DNA, we employed a read mapping approach. The short-sequence reads used to generate the *Pr. xenopodis* assembly [40] data were downloaded from the Sequence Read Archive (SRA) (BioProject accession number PRJEB2979). The reads were trimmed for adapters and low-quality sequence regions using Trimmomatic v.0.39 [47] and subsequently mapped to the *X. laevis* genome assembly (accession number GCF_017654675.1) using BWA v.0.7.17 [48]. Basic alignment statistics were obtained using Samtools v.1.19 [49], and sequence coverage of the mapped reads was evaluated via BedTools2 v.2.31.1 [50] using *genomecov –bga* options. Read coverage along the *X. laevis* genome was plotted using the karyoplotteR [51] R package v.1.8.4. Statistical analysis was employed to determine whether there was a significant difference in the density of mapped reads along 1 Mb windows of the *X. laevis* genome using a Poisson test (significant threshold ≤ 0.01) (script deposited in: https://github.com/kalebgatto/R_codes/main/plot_coverage.R) in the software R v.4.4.1 [52]. Regions in the *X. laevis* genome that presented significantly high numbers of mapped reads were inspected for (a) the presence of 5S rDNA repeats via BLASTn v.2.16.0, (b) other classes/families of repetitive DNA employing RepeatMasker v.4.1.7 [53], and (c) protein-coding genes by visualization in NCBI Genome Data Viewer v.3.49 [54]. If *Pr. xenopodis* sequence libraries have high contamination profiles from *X. laevis*, similar coverage values would be expected for all repetitive DNAs.

## 3. Results

### 3.1. Incongruities in the ML Analysis

The maximum likelihood analysis of all the presumed and confirmed 5S rRNA gene sequences of anurans, flatworms, and nematodes revealed incongruities between the inferred sequence groups and the phylogenetic relationships of the sampled species. One of the incongruities refers to the worm *Protopolystoma xenopodis*. In the ML dendrogram, three of the four sequences of *Pr. xenopodis* were grouped together with those of the remaining flatworms, whereas the other sequence (isolated from contig0184163 of the *Pr. xenopodis* genome assembly) clustered with the anuran 5S rRNA genes, along with sequences of *Xenopus laevis* and *X. borealis* (Figure 1 and Appendix A). This latter *Pr. xenopodis* sequence was highly similar (98.03%) to the somatic type of the 5S rRNA gene sequence of *X. laevis* (J01009, M35055 and X12622) and was 98.30% similar to the somatic type of the 5S rRNA gene of *X. borealis* (K01537 and V01426). Compared with the oocyte type of the 5S rRNA gene of *X. laevis* and *X. borealis*, the sequence of *Pr. xenopodis* showed 94.14% and 92.22% similarity, respectively. In contrast, the 5S rRNA gene sequence of *Pr. xenopodis* was only 67.70% similar to the other type of 5S rRNA gene found in the genome assembly of this species (Table 1).

Another incongruity in the ML analysis refers to the type I 5S rDNA sequences of *Pseudis fusca* and *Ps. tocantins* species, which were nested among the 5S rDNA sequences of the nematode species *Bursaphelenchus xylophilus* and *Subanguina moxae* (Figure 1). When the presumed transcribed region of these type I 5S rDNA sequences of *Ps. tocantins* and *Ps. fusca* was compared to the type II 5S rDNA of *Ps. tocantins* and *Lysapsus limellum*, type I 5S rDNA of *Ps. bolbodactyla*, and the nematode sequences, they showed 64.14%, 79.61%, and 84.57% similarity, respectively. The mean similarity of the anuran sequences that clustered together in the same group in the ML dendrogram (blue-shaded rectangle in Figure 1) was 79.67%, a value similar to those found for the flatworm sequences (sequence from contig0184163 of *Pr. xenopodis* excluded) and for the nematode sequences (Table 1).

### 3.2. An Extended Comparison Between the 5S rDNA of Pr. xenopodis and Xenopus Species

To expand the analysis of the sequence found in *Pr. xenopodis* and the 5S rDNA of the *Xenopus* species, we analyzed the regions that flanked the 5S rRNA gene in the *Pr. xenopodis* genome assembly. BLAST searches using the somatic type of the 5S rRNA gene sequence of *X. laevis* (J01009) as a query returned three contigs of the *Pr. xenopodis* genome assembly (Table 2), and all of them had sequences annotated as the 5S rRNA gene (Figure 2). However, only contig0184163 showed both high similarity and high query coverage in this analysis (Figure 2A). When the oocyte type of the 5S rRNA gene of *X. laevis* was used as a query, the BLAST searches in the *Pr. xenopodis* genome assembly identified the same three contigs found during the former analysis, but the similarities were restricted to part of the 5S rRNA gene (Table 2; Figure 2).

When contig0184163 of *Pr. xenopodis* was aligned with anuran 5S rDNA sequences, we noted that the similarity between the 5S rDNA sequence of *Pr. xenopodis* and the somatic type of 5S rDNA of *X. laevis* was not restricted to the 5S rRNA gene but also extended to the NTS (Figure 3). *Pr. xenopodis* contig0184163 showed a 75.11% overall similarity with the *X. laevis* somatic 5S rDNA (5S rRNA gene + NTS; J01009). While these sequences were identical in the 5S rRNA gene region, their NTSs were 71.28% similar. With respect to the somatic type of 5S rDNA of *X. borealis*, a lower level of similarity with contig0184163 of *Pr. xenopodis* was found (Figure 3). Although these sequences were highly similar (98.30%) with respect to the 5S rRNA gene, their NTSs were only 59.50% similar to each other. In contrast, the NTS of the 5S rDNA oocyte type of the *Xenopus* species and the NTS of the 5S rDNA sequence in the contig0184163 of *Pr. xenopodis* were highly distinct (Figure 3).

### 3.3. Assessing the Hypothesis of Frog DNA Contamination in the Pr. xenopodis Genome Assembly

When we used single-copy and multigene family sequences from *X. laevis* as queries in BLAST searches against the *Pr. xenopodis* genome assembly, we did not recover any significant similarities (Appendix A). In the case of the 40S rDNA intergenic spacer (IGS), sequence X05264 (which contains not only the IGS but also the final portion of the 28S gene) corresponded with only small segments of the promoters and enhancer elements in *Pr. xenopodis* contig0182281 (Appendix A). BLAST searches using *Pr. xenopodis* contig0182281 as a query against the *X. laevis* nucleotide collection of NCBI detected two 40S rDNA cloned sequences (X05264 and X02995, the latter of which contains the transcribed region of the precursor rRNA). However, the similarity with the *Pr. xenopodis* sequence was restricted to regulatory or transcribed regions, i.e., the 3′-end of the second enhancer, the promoter region, and the external transcribed spacer (Appendix A). Although a BLAST search using a cloned segment of *X. laevis* rDNA containing 18S-5.8S-28S rRNA genes and the internal transcribed spacers revealed correspondence in several contigs of *Pr. xenopodis*, the majority of the hits had a low alignment size (less than 100 bp) (Appendix A).

BLAST searches using the histone gene sequences of *X. laevis* as queries revealed positive hits in the coding region of the highly conserved H3, H4, H2B, and H2A genes, whereas the H1 gene of *Pr. xenopodis* was not recovered in this analysis (Appendix A). In addition, the spacer DNA between each histone gene of *X. laevis* showed only small segments of alignment with high e-values in the *Pr. xenopodis* genome assembly (Appendix A).

BLAST searches using both high- and low-copy-number TEs from *X. laevis* revealed limited correspondence in the *Pr. xenopodis* genome assembly. The contigs with positive hits displayed low query cover and alignment sizes in the contigs with positive hits (Appendix A). Notably, the BLAST search for the highly abundant Tc1-Mariner elements in the *X. laevis* genome did not yield any hits in this analysis.

The read mapping approach resulted in 8251826 and 35565694 mapped reads along the *X. laevis* genome, accounting for approximately 7.55% and 20.41% of the total number of reads from the *Pr. xenopodis* short-read libraries. Notably, these two different libraries yielded similar read mapping density profiles (Figure 4). Only five 1 Mb windows along the *X. laevis* genome presented a significantly high density of mapped reads from both libraries (Figure 4 and Appendix A). The first one of these windows is on chromosome 6L and contains 5S rDNA clusters (Appendix A). BLAST searches using NTS sequences of the *X. laevis* somatic 5S rDNA revealed the presence of somatic 5S rDNA in this region (Figure 5). Additionally, BLAST searches using the NTS of oocyte-specific 5S rDNA also revealed significant hits. However, the oocyte-specific 5S rDNA NTS contains repetitive segments consisting of a variable number of short repeats, which accounts for these results (Appendix A). BLAST searches using these short repeats as queries revealed high contents of these repetitive sequences in the *Pr. xenopodis* genome (Appendix A). The second region significantly mapped by *Pr. xenopodis* reads is on chromosome 7S and also contains oocyte-specific 5S rDNA (Appendix A). Finally, the third region with a significant number of mapped reads is located between positions 20 Mb and 30 Mb on chromosome 3L of *X. laevis* and is annotated for 18S-5.8S-28S rRNA genes (Figure 4 and Appendix A). It is worth noting that the *Pr. xenopodis* reads aligned with the transcribing region of this rDNA. In addition, BLAST searches using the intergenic spacer of the 40S rDNA of *X. laevis* as a query revealed only one contig of the *Pr. xenopodis* genome with significant hits, but the correspondence between the *X. laevis* and *Pr. xenopodis* sequences was limited to regulatory regions (promoter and enhancer) and the external transcribed spacer.

In addition to the 5S and 18S-5.8S-28S rDNA segments, regions with a significant number of mapped reads were found on chromosomes 7S and 8S (Figure 4). RepeatMasker revealed that these regions are highly populated by microsatellite motifs and/or telomeric repeats (Appendix A).

### 3.4. Comparison of the Type I 5S rDNA of Pseudis tocantins and Pseudis fusca with the 5S rDNA of Bursaphelenchus xilophilus and Subanguina moxae

The BLAST searches using the type I 5S rDNA of *Ps. tocantins* (KX170899) as a query against the genome assemblies of *B. xylophilus* identified several contigs annotated as the 5S rRNA gene, with the majority of the hits being found in tandem arrays (Appendix A). The similarity of the presumed transcribed region of the type I 5S rDNA of *Ps. tocantins*/*Ps. fusca* with the 5S rRNA gene of *B. xylophilus* was 85.33%, while the similarity with *S. moxae* was 88.83%. The NTS of type I 5S rDNA of *Ps. tocantins* and *Ps. fusca* was not similar to any of the sequences currently available in public databases.

## 4. Discussion

### 4.1. Evidence of a Possible HGT Between Frogs and Worm Parasites

In the present work, we found evidence of possible 5S rDNA HGT between frogs and parasite worms. The most noticeable case refers to the frog genus *Xenopus* and its parasite, *Protopolystoma xenopodis*. The 5S rDNA from *Xenopus* was first characterized by Brown et al. [55] and Brown and Sugimoto [56]. In *X. laevis,* there are two different types of 5S rDNA: (a) a somatic type, which is transcribed in all somatic tissues, with an NTS of 768 bp; and (b) an oocyte-specific type, which is transcribed only in oocytes, with an NTS that varies from 537 bp to 701 bp [57,58]. A similar situation has also been observed for *X. borealis* and *X. tropicalis* [58,59].

The genome assembly of the parasite *Pr. xenopodis* has a contig (PXEA_contig0181163) that is highly similar to the somatic type of 5S rDNA of *X. laevis*, both in the gene region and NTS (Figure 3), and it greatly differs from the other 5S rRNA genes annotated to this worm species. In the ML analysis, this 5S rRNA gene sequence of *Pr. xenopodis* clustered together with the somatic 5S rRNA gene sequences of *X. laevis* (J01009) and not in the cluster of flatworm sequences (which included the 5S rRNA gene sequences from other contigs of *Pr. xenopodis*) (Figure 1).

The phylogenetic incongruity between a gene dendrogram and a species tree is the first indicator of HGT events, particularly when it involves phylogenetically distant taxa [4,60]. Despite the high conservation of the 5S rRNA gene across all metazoans, which is attributed to strong selective pressure and the action of homogenizing mechanisms [61,62], vertebrate sequences of the 5S rRNA gene cluster separately from those of other animals, such as nematodes and flatworms [33]. Therefore, our findings raise the hypothesis that HGT occurred from *X. laevis* to a parasitic worm.

The inference of HGT events between host and parasite species is sometimes controversial, primarily owing to genuine concerns about DNA sample contamination [4,63,64]. *Protopolystoma xenopodis* infests the urinary system of *Xenopus* species and typically feeds on their blood [65,66,67]. The samples of *Pr. xenopodis* used for genome sequencing were collected from wild-caught *Xenopus laevis* (as described in Appendix A of Coghlan et al. [40]). Hence, the presence of contaminant DNA from *Xenopus laevis* in the libraries employed for sequencing the genome of *Pr. xenopodis* is a possibility and deserves attention. To assess this possibility, we first conducted a thorough search within the genome assembly of *Pr. xenopodis* for sequences exhibiting high similarity to those of *X. laevis*, which could indicate the presence of contaminant *Xenopus* DNA. Our BLAST searches revealed only H3, H4, H2B, and H2A histone genes and 18S-5.8S-28S rDNA (Appendix A), which are highly conserved genes among eukaryotes [68,69]. Notably, in the case of histone gene clusters, significant alignment was not observed for the H1 gene, which is the least conserved histone among eukaryotes [68]. In addition, the spacer DNA between each histone gene of *X. laevis* presented only small segments of alignments with high e-values in the *Pr. xenopodis* genome assembly. No single-copy gene that is supposedly shared between *X. laevis* and *Pr. xenopodis* was identified (Appendix A). The search for transposable elements also failed to identify any significant correspondence between *X. laevis* and the *Pr. xenopodis* genome assembly (Appendix A). Notably, even for the Tc1-Mariner element, which is highly abundant in the *X. laevis* genome [70], no evidence of sharing was found.

The second approach we used to investigate the hypothesis of contamination in the *Pr. xenopodis* genome assembly involved the mapping of *Pr. xenopodis* reads onto the *X. laevis* genome assembly, and it also failed to provide supporting evidence in this context. Approximately 7.5 and 20.4% of the two available *Pr. xenopodis* short-read libraries were mapped onto the *X. laevis* genome, and both libraries, which were made from different sources and had different total numbers of reads, yielded peaks of read density in the same regions of the *X. laevis* genome assembly. All the peaks of the mapped reads are located in regions enriched for repetitive and highly conserved sequences (i.e., rRNA and microsatellite motifs). One of these regions with a high density of mapped reads corresponded to 40S rDNA, which was also one of the multigene families identified in our abovementioned BLAST search strategy. Previous studies estimated that there are 400–600 copies of the 40S rDNA unit in the haploid genome of *X. laevis* [71,72,73,74]. Therefore, contamination of this type of sequence could be plausible. However, if contamination had indeed occurred, we would expect to recover not only the highly conserved 18S-5.8S-28S rRNA genes but also the intergenic spacer, which is highly variable among distantly related species [69,75,76,77]. In contrast, the region of the *X. laevis* genome corresponding to 40S rDNA presented a high density of *Pr. xenopodis* reads aligns with its transcribing region. In BLAST searches using the intergenic spacer of the 40S rDNA of *X. laevis* as a query, only one contig of the *Pr. xenopodis* genome was retrieved, but the correspondence between the *X. laevis* and *Pr. xenopodis* sequences were limited to regulatory regions (promoters and enhancers) and external transcribed spacers (Appendix A).

A third piece of evidence that undermines the contamination hypothesis arises from the failure to detect the oocyte-specific type of *X. laevis* 5S rDNA in the *Pr. xenopodis* genome assembly. According to Peterson et al. [58], oocyte-specific 5S rDNA is much more abundant than somatic 5S rDNA in the *X. laevis* genome. In the haploid genome of *X. laevis*, there are approximately 20,000 and 1300 copies of major and minor oocyte type sequences, respectively, and approximately 400 copies of the somatic 5S rDNA unit [58]. In our analyses of the *Pr. xenopodis* genome assembly, we found evidence for the presence of a sequence that was highly similar to the somatic 5S rDNA of *X. laevis*, which is the least abundant type of 5S rDNA in the *Xenopus* genome, but we observed no evidence for the presence of the most abundant 5S rDNA type. These findings contradict expectations in a scenario of DNA contamination.

Lastly, the 5S NTS found in the contig PXEA_0181163 of *Pr. xenopodis* shows some differences in relation to the sequence of *X. laevis*, which would not be expected in the hypothetical scenario of contamination of the worm libraries with *X. laevis* DNA. Therefore, we found no evidence to support the hypothesis that contamination of the *Pr. xenopodis* genome assembly with *X. laevis* DNA could explain the remarkable similarities observed between the 5S rDNA from these distantly related taxa.

In addition to DNA sample contamination, another alternative explanation to HGT that we should consider is convergent evolution. However, this hypothesis appears unlikely in the case we discuss here, as we observed extensive similarity in the NTS of a 5S rDNA sequence of *Pr. xenopodis* and the somatic 5S rDNA of *X. laevis*. Although a regulatory role in gene transcription has been attributed to the NTS of 5S rDNA, several studies have shown that only small portions of the NTS are functionally relevant in this context [78,79,80], with the major transcriptional control region—the internal control region (ICR)—located within the transcribed portion of the gene itself [81]. As a result, a low adaptive value is expected for NTS, which aligns with empirical observations of high variability in both size and nucleotide composition of NTSs, even within a single genome [33,41,82]. Therefore, the extensive similarity found between the NTS of a 5S rDNA sequence of *Pr. xenopodis* and that of the somatic type of 5S rDNA of *X. laevis* (Figure 3) does not support the hypothesis of convergent evolution. 

Apart from discarding these alternative hypotheses, another important point to be considered when evaluating potential HGT is the possibility of alien DNA being incorporated into the host genome and transmitted to offspring. Some elements have been frequently evoked to explain the incorporation of foreign DNA, including TEs, extracellular vesicles, tunneling nanotubes, viral transduction, and circulating cell-free DNA [4,83,84]. In parallel, ecological relationships, such as endosymbiosis and parasitism, are widely recognized as contexts that can facilitate HGT, given the intimate and often prolonged contact between organisms [4,32]. Nevertheless, identifying molecular footprints of HGT remains extremely challenging, and, as noted by Keeling [84], most reported cases of HGT in eukaryotes have not elucidated the precise mechanisms by which foreign DNA was acquired. In the case we examine here, we were also unable to identify any molecular footprint of the hypothetical transfer of alien DNA to the flatworm genome, but we highlight some points to support the plausibility of such an event, emphasizing that the conditions and biological interactions involved could feasibly allow for it. The first one is that larvae and adult individuals of *Pr. xenopodis* feed on *X. laevis* blood [65,66], which may have favored the potential transfer of DNA from the frog to the flatworm. This aligns with the widely accepted hypothesis known as “you are what you eat”, which posits that dietary DNA may be incorporated into genomes [64,85]. Additionally, it is worth noting that in adult *Pr. xenopodis*, the testes and ovaries are located near the buccal apparatus, and the genito-intestinal canal connects the oviduct to the intestines [65], which may facilitate the transmission of incorporated foreign DNA to germ cells. Alternatively, we should also consider the hypothesis that free DNA from *X. laevis* present in its urine could be transferred to *Pr. xenopodis* eggs since these eggs are released immediately into the frog’s urinary bladder shortly after formation, because this flatworm has no uterus [66,67]. In this latter scenario, the flatworm egg would represent a weakly protected stage for foreign DNA entry, aligning with the weak-link model proposed by Huang [86] to explain HGT in eukaryotes.

In conclusion, we did not find any evidence that refutes the HGT hypothesis as the most plausible explanation for the discovery of a 5S rRNA gene identical to that of *X. laevis* in the genome of *Pr. xenopodis*. In an HGT scenario, the few differences found between the NTS of the somatic 5S rDNA of *X. laevis* and the contig0181163 of *Pr. xenopodis* could be explained by sequence divergence following the hypothetical HGT event, suggesting that this gene transfer may have occurred some time ago. Alternatively, some species closely related to *X. laevis*, for which there is no information concerning the 5S rDNA sequence, may have been the donor species in the supposed HGT event. Therefore, further analyses of 5S rDNA sequences of *Xenopus* species, such as *X. petersii*, *X. poweri*, and *X. gilli*, among others, as well as other species of *Protopolystoma* that infect *Xenopus* species [87] may be helpful in future studies.

In addition to the abovementioned case, another noteworthy finding of our analyses was the clustering of the type I 5S rDNA from the anuran genus *Pseudis* within the nematode group in the ML analysis of 5S rRNA genes. The 5S rRNA gene sequence identified as the most similar to the type I 5S rDNA of *Ps. tocantins* and *Ps. fusca* grouped together with *Bursaphelenchus xylophilus* and *Subanguina moxae* 5S rRNA gene sequences in a clade composed of nematode sequences. Curiously, the NTS of type I 5S rDNA of *Pseudis* is not similar to any of the 5S rDNA sequences currently available in public databases. The nematodes *B. xylophilus* and *S. moxae* are species from the Rhabditida order (Tylenchomorpha infraorder) and they infect plant species [88,89], and it seems odd to find 5S rDNA sequences similar to those nematodes in anuran species. However, some nematode parasites that infest *Ps. paradoxa* (a species closely related to *Ps. fusca* and *Ps. tocantins*), such as species of *Cosmocerca*, *Gyrinicola*, *Rhabdias*, and *Spiroxys*, are also from the Rhabditida order [90,91]. Accordingly, HGT may have occurred from some nematode species to a *Pseudis* frog, and because the available database for nematode 5S rDNA is scarce, we were unable to identify any correspondence between the *Pseudis* 5S rDNA NTS and any known 5S rDNA sequence from nematode species.

However, two alternative hypotheses may explain the clustering of the type I 5S rRNA gene sequence of *Pseudis* together with the nematode sequences: (a) contamination of the *Pseudis* DNA samples used to isolate 5S rDNA sequences with nematode DNA and (b) error in the ML inferences of 5S rRNA gene relationships. Because the sequences classified as type I 5S rDNA of *Pseudis* were isolated from two different species, i.e., *Ps. tocantins* [46] and *Ps. fusca* [41], the hypothesis that nematode contaminant DNA occurred among the 5S rDNA of *Pseudis* species is weakened. The ML analysis that nested the type I 5S rDNA of *Pseudis* among the nematode sequences was based on the 5S rRNA gene region, which is approximately 120 bp in length. Therefore, although HGT from a nematode to a *Pseudis* frog is a possible hypothesis, the identification of similarities in the 5S NTS would also be very helpful in assessing this possibility in future studies.

### 4.2. HGT and the Diversity of 5S rDNA

The average similarity reported in the literature for the 5S rRNA gene of anurans is approximately 84% [41], which is consistent with the mean value we found here (80%). Interestingly, 5S rRNA gene similarity among flatworm species, as well as among nematode species, is similar to that reported for anurans, whereas the similarity values between Anura and flatworms or nematodes are less than 70%. The variability in the 5S rDNA NTS, in contrast, is pronounced, with the NTS varying in both sequence composition and length among closely related species and even within the same species [33,38,41,92]. In this sense, similarity in small regions of the NTS from distantly related species may indicate a significant role for such regions, potentially contributing to their evolutionary conservation, as previously discussed by Vierna et al. [33] and Targueta et al. [41]. In contrast, extensive similarity between NTSs from distantly related taxa is not expected and might be consistent with HGT hypotheses. In a comprehensive study of the 5S rDNA of metazoans, Vierna et al. [33] proposed HGT as one possible explanation for the high similarity found among 5S rDNA NTSs from distantly related taxa, such as the porifer *Reniera* sp. and the mollusk *Lottia gigantea*. The case we reported here, concerning the NTS of the *X. laevis* somatic 5S rDNA and the NTS of the *Pr. xenopodis* (contig0181163), is consistent with this perspective.

The evolution of in tandem repetitive sequences, such as 5S rDNA, involves two important processes, which are not mutually exclusive: concerted evolution and birth-and-death evolution. The concerted evolution model explains that a mutation in a monomer may spread to others by non-reciprocal recombination, resulting in the homogenization of DNA sequences [62,93]. Moreover, the birth-and-death model refers to the creation of new sequence variants and their maintenance or elimination owing to natural selection or genetic drift [94,95,96]. Both models have been observed to act together in various genomes [82,96,97,98,99]. In addition, regarding the 5S rDNA evolution of anurans, recombination with the satellite DNA PcP190 should be considered [41]. This satellite DNA originated from 5S rDNA and is widespread in Hyloidea [41,100]. Occasional events between 5S rDNA and PcP190 satellite DNA were supported by the finding of a chimeric fragment in the genome of one species of *Lysapsus* [41], suggesting the hypothesis of interchange between these repetitive DNA families.

In this scenario of 5S rDNA evolution, HGT events emerge as additional sources of new variants, alongside mutation and intraspecific recombination. Vierna et al. [33] had previously hypothesized HGT as a possible mechanism involved in the evolution of 5S rDNA in metazoans, and here we contribute to this discussion by presenting specific candidate cases involving anurans. However, whether horizontally transferred 5S rDNA is effectively transcribed is still an open question to be further investigated.

## 5. Conclusions

We revealed candidate cases of cross-phylum HGT involving frogs and worms, highlighting the possible contribution of HGT to the high diversity observed in the 5S rDNA family.

## Figures and Tables

**Figure 1 biomolecules-15-01001-f001:**
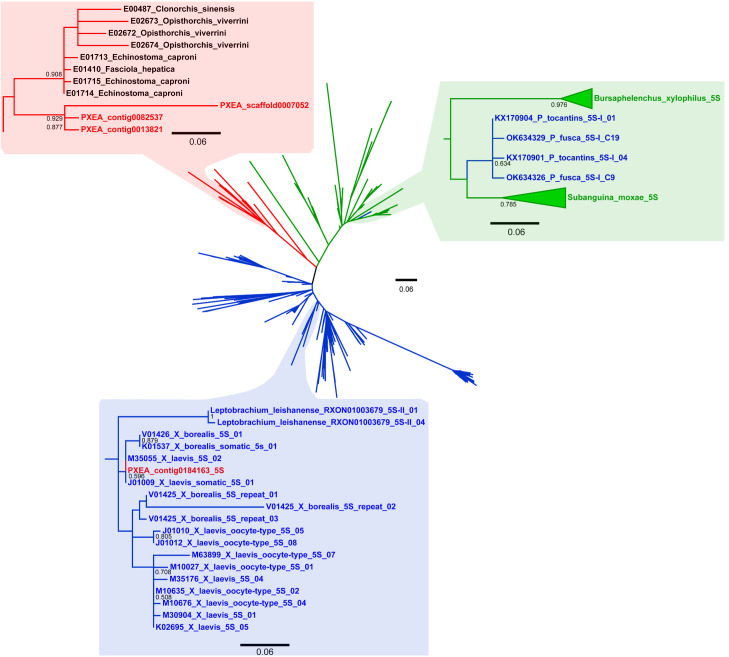
Maximum likelihood analysis of the 5S rRNA gene region from anurans (blue), flatworms (red), and nematodes (green). The blue-shaded rectangle highlighted from the radial dendrogram shows the relationship of one sequence of *Protopolystoma xenopodis* (PXEA_contig0184163) with sequences of *Xenopus*. The green-shaded rectangle highlighted from the radial dendrogram shows the relationship of the 5S rDNA type I sequences from *Pseudis tocantins* and *Pseudis fusca* with the sequences from the nematode species *Bursaphelenchus xylophilus* and *Subanguina moxae*.

**Figure 2 biomolecules-15-01001-f002:**
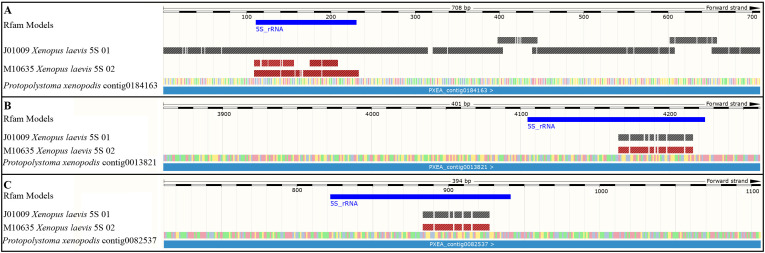
Three contigs retrieved from the genome assembly of *Protopolystoma xenopodis* aligned with two 5S rDNA sequences of *Xenopus laevis* (J01009 and M10635). (**A**–**C**). Contigs 0184163, 0013821, and 0082573 of the genome assembly of *Protopolystoma xenopodis*, respectively. The blue lines identify regions annotated as 5S rRNA genes based on Rfam models. In A, the two segments of the M10635 sequence aligned with the 5S rRNA gene of *Pr. xenopodis* refer to a 5S rRNA pseudogene of *X. laevis*. These figures were generated from GenomeBrowser via the BLAST tool search in WormBase Parasite.

**Figure 3 biomolecules-15-01001-f003:**
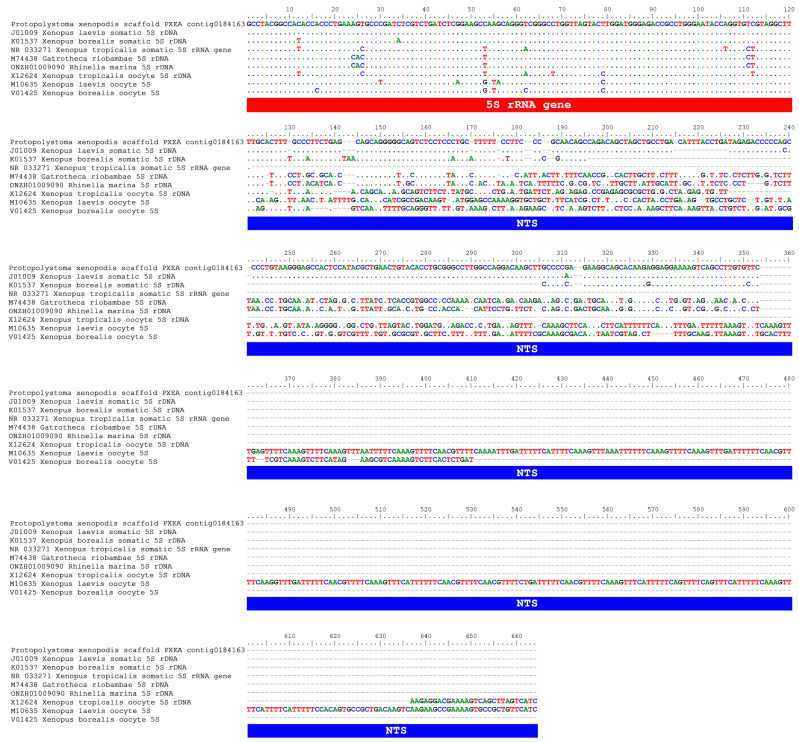
Alignment of 5S rDNA sequences from *Xenopus laevis* (somatic and oocyte types), *X. borealis* (somatic and oocyte types), *X. tropicalis* (somatic and oocyte types), *Gastrotheca riobambae*, *Rhinella marina*, and *Protopolystoma xenopodis* (contig0184163 of the genome assembly GCA_900617795). Note that the 5S rDNA sequence of *Pr. xenopodis* is very similar to the somatic type of 5S rDNA of *X. laevis*, and the similarity between these sequences is not restricted to the gene region (in red) but also extends to the NTS (in blue).

**Figure 4 biomolecules-15-01001-f004:**
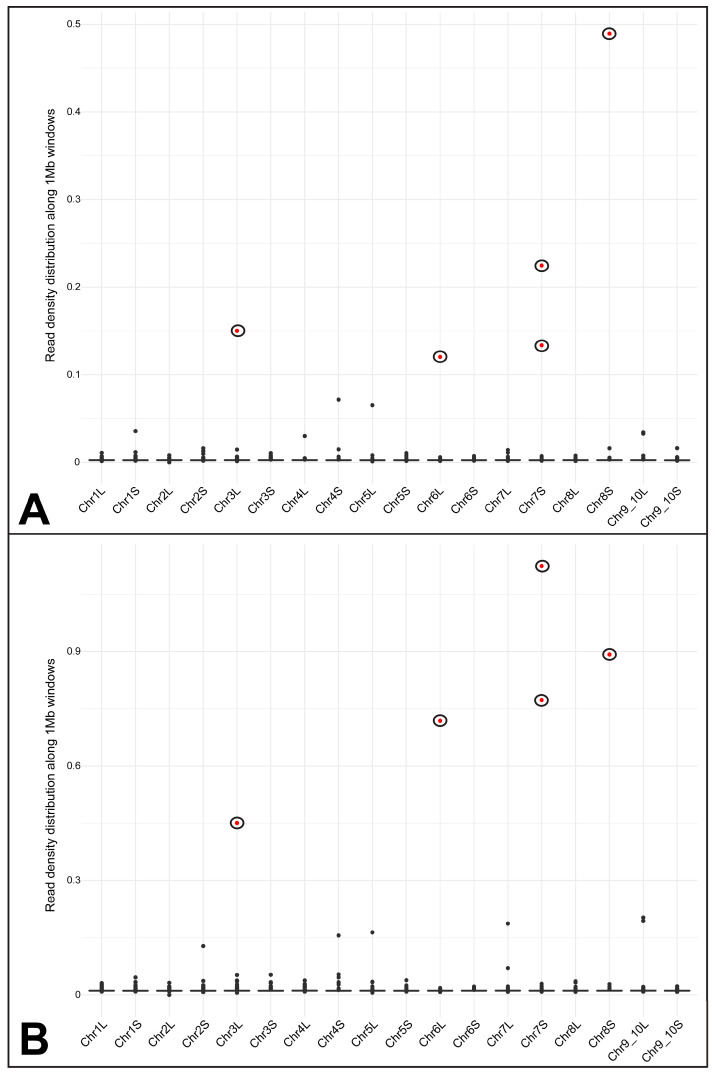
Read mapping densities of *Protopolystoma xenopodis* ERR065030 (**A**) and ERR304767 (**B**) libraries along 1 Mb windows of the *X. laevis* chromosomes. The circled dots indicate the windows that presented a significant number of mapped reads according to the Poisson test (*p* value ≤ 0.01) in both cases.

**Figure 5 biomolecules-15-01001-f005:**
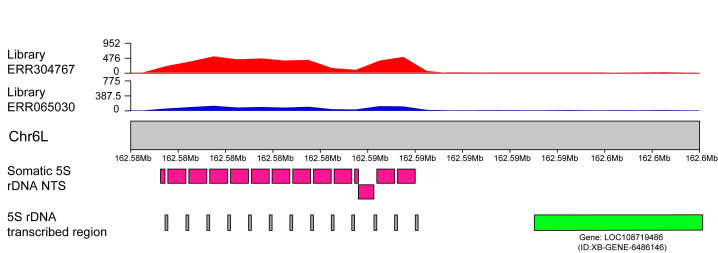
Mapping of reads from two *Protopolystoma xenopodis* short-read libraries (ERR3044767 and ERR065030) to a region of *Xenopus laevis* chromosome 6L that harbors a cluster of the somatic type of 5S rDNA. On the X axis, chromosome positions are indicated in megabases (Mb), and the Y axis shows the read mapping densities. The bottom images show the BLAST alignment results of the NTS of somatic 5S rDNA (pink) and the annotated 5S rRNA gene sequences (gray). The gene LOC108719486, which is located near this 5S rDNA cluster, is shown in green.

**Table 1 biomolecules-15-01001-t001:** Similarity (%) among the presumed transcribed regions of the 5S rDNA and 5S rRNA genes from the anurans, flatworms, and nematode haplotypes included in the present study. The similarity values within each group are highlighted in gray on the diagonal. * Sequences obtained from the scaffolds/contigs 0082537, 0013821, and 00070052 of the genome assembly of *Protopolystoma xenopodis*.

	1	2	3	4	5	6
1. Anura	80.00					
2. Platyhelminthes	67.94	85.50				
3. Nematoda	68.45	68.73	82.35			
4. *Pseudis* 5S-I	70.16	68.14	84.57	98.03		
5. *Pr. xenopodis* contig-0184163	84.55	73.51	75.29	81.91	-	
6. *Pr. xenopodis* 5S *	63.37	80.27	64.24	64.58	67.70	86.83

**Table 2 biomolecules-15-01001-t002:** Results of BLAST searches for two types of 5S rDNA from *Xenopus laevis* (J01009 and M10635) in the *Protopolystoma xenopodis* genome assembly.

5S rDNA (Query)	*P. xenopodis* Contigs	Annotation	Query Cover (%)	E-Value
*Xenopus laevis* 5S (J01009)	contig0184163	rRNA 5S	82	5 × 10^−139^
	contig0082537	rRNA 5S	6	7.4 × 10^−6^
	contig0013821	rRNA 5S	2	1.2 × 10^−4^
*Xenopus laevis* 5S (M10635)	contig0184163	rRNA 5S	31	5 × 10^−49^
	contig0082537	rRNA 5S	6	0.0013
	contig0013821	rRNA 5S	2	0.021

## Data Availability

All the data generated or analyzed during this study are included in this published article and its Appendix A.

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
