# Peer review of "Could Horizontal Gene Transfer Explain 5S rDNA Similarities Between Frogs and Worm Parasites?"

_biomolecules, 2025, doi:10.3390/biom15071001_

Round 1

Reviewer 1 Report

Comments and Suggestions for Authors

The MS. by K.P. Gatto et al. entitled “Horizontal Gene Transfer between Frogs and Worm Parasites: a Tale from 5S rDNA“ presents interesting data obtained by the analysis of the publicly available databases. The authors discovered a peculiar similarity between 5S rDNA of Xenopus laevis and its parasite the Monogenea worm Protopolystoma xenopodis. Remarkably, the similarity was observed not only in the transcribed 120 bp regions, but also partly in the spacer. The title of the work contains a claim that the Horisontal Gene Transfer (HGT) actually took place between the frog and the worm. Moreover, it suggested a similar transfer from another amhibian (Pseudis) to nematodes. Apparently, the authors believe that HGT between remote species of animals is a common phenomenon.

Now extraordinary claims require very convincing justifications. The authors went to great lengths to exclude the possibility of the interspecies contamination in the samples of the parasite worm. But the MS. contains no further arguments in favour of the HGT hypothesis. The authors did not pay due attention to the following difficulties.

Firstly, how could the brachian DNA penetrate the eggs of the worm? The authors write (page12, line 413 – page 13, line 414): “…we may hypothesize that free DNA from X. laevis present in its urine could be transferred to Pr. xenopodis eggs…“ I think, the possibility cannot be excluded altogether, but the suggested process does not seem very likely. For one thing, the concentration of extracellular DNA in the urine would be very low, and the DNA could hardly penetrate the intact monogenean eggs, which are known to be impervious to medical treatment.

Secondly, if the DNA did penetrate, how could it incorporate into the genome? As far as I know, in the well proven cases of incorporation (both in the nature and in the laboratory), the donor’s nucleic acid should be equipped with special sequencies. But the authors of the MS. either did not seek for or sought but did not find any symptoms of transposition. Could a non-specified recombination be the answer? We have to observe that the extraordinary similarity between X. laevis and Pr. xenopodis referred to short sequences containing 120 bp gene and only a part of the spacer, but no whole clasters. One would expect longer sequences for recombination. In page 14, lines 481-482, the authors write: “In addition, regarding the 5S rDNA evolution of anurans, recombination with the satellite DNA PcP190 should be considered.“ But, as I understand, that recombination with only 190 bp long repeat is also just hypothetical. At any rate, the authors of the MS mentioned no symptoms of such recombination (e.g. presence of the sequences from “hypervariable region“ of PcP190) or any other processes ensuring the integration of the frog DNA into Pr. xenopodis genome.

Thirdly, if the exogeneous sequences could find their way into the worm’s genome, why should they be settled there and not be blocked or deleted? Why should  they propagate in the population (and eventually in the entire species) to the exclusion of the host’s variants? Do they confer the selective value comparable to that of the drug resistence plasmids?  

Fourthly, HGT in the repetitive genes, such as 5S rDNA, would be still more problematic, since the new variant should supplant not just one unit, but all the clusters, perhaps situated on different chromosomes. Here I would observe that the MS. has to provide an additional information about the organization of 5S rDNA in the studied parasitic worms. A simple  figure would do showing the number of copies, the number of clusters, the number of chromosomes carrying these clusters, the situation of 5S rDNA in relation to other ribosomal genes (for they may be located on different chromosomes or in the same locus). All this data are relevant to the case.

Finally, no mention is made in the MS. of what might be a more plausible alternative to the HGT hypothesis, namely, similarity of sequence determined by the similarity of function. Thus animals employing echolocation have not only analogous physiolological features but also remarkable similarity of the DNA sequences engaged in the process. Should we discuss a gene transfer between bats and dolphins? Or should we better look for the analogous functions (perhaps not ribosomal) of the similar NTS sequences?

A few minor points:

- Page 9, line 257-258. “…rDNA cloned sequences (X05264 and X02995, the latter of which contains the transcribed region of the precursor rRNA).“

X02995 is missing in Table S4, which shows alignment similarity

- Page 9, line 259. “…the 5’-end of the second enhancer…“

It is actually 3‘ end.

- Page 9, line 291. “A third region with a significant amount of mapped reads.“

It is not quite clear what are the first two.

- Fig. 5. What are the numbers at the ordinate?

- Discussion. For the convenience of the reader, it would be advisable to supply references to the figures and tables throughout the Discussion.

- Page 13, lines 419-420. “In conclusion, we cannot find any evidence refuting the HGT hypothesis to explain the discovery of a 5S rRNA gene identical to that of X. laevis in the genome of Pr. xenopodis“

Two different ideas seem to be confused in this sentence: plausibility of HGT hypothesis and finding an alternative hypothesis to explain the data.

- Page 13, line 421. “According to this scenario,…“

It is not clear what this phrase refers to, for the word “scenario“ was mentioned previously in reference to the contamination.

To conclude, if the authors insist on the claim in the title of the MS., I would suggest a major revision that would provide convincing facts in support of the HGT hypothesis. Otherwise, I think the authors could just change the title (e.g. by using question mark at the end would), expound the above-mentioned difficulties attending the HGT hypothesis, and discuss the alternative interpretation of the data. The MS. thus would be no less interesting, but more correct.

Author Response

Response to reviewer #1

We thank the Reviewer for the analysis and helpful suggestions on our MS. We have made changes in the revised manuscript to enhance clarity. We answer below each comment of the Reviewer.

Comment #1: Apparently, the authors believe that HGT between remote species of animals is a common phenomenon.

Answer: In our MS, we propose that HGT is a probable explanation for the high similarity observed between 5S rDNA of frogs and parasites, and we provide a point-by-point analysis of several factors. At no point in our MS do we assume or suggest that HGT is a common phenomenon, even though there are robust evidence in the literature for HGT, as cited in the introduction (lines 57-66).

Comment #2: Now extraordinary claims require very convincing justifications. The authors went to great lengths to exclude the possibility of the interspecies contamination in the samples of the parasite worm. But the MS. contains no further arguments in favour of the HGT hypothesis.

Answer: We totally agree that proving HGT is a very hard task. In our MS, we intended to present a detailed analyses of the HGT hypothesis and the alternative hypothesis of contamination, aiming to propose that HGT is a plausible scenario to explain the presence of Xenopus 5S rDNA among Protopolystoma xenopodis genome assembly, although definite evidence is still needed. In the revised MS, to make it clearer, we rephrased several sentences throughout the main text, including the title. Additionally, we also included an analysis of the hypothesis that consider convergent evolution as an explanation for the data we observed and discussed the reasons why it is not a likely hypothesis. Given that Xenopus laevis is a model organism and one of the few frogs for which a high-quality chromosome-level genome assembly is available, we believe that our suspicion of HGT involving 5S rDNA is an important finding and should be shared with the research community.   

Comment #3: The authors did not pay due attention to the following difficulties. Firstly, how could the brachian DNA penetrate the eggs of the worm? The authors write (page12, line 413 – page 13, line 414): “…we may hypothesize that free DNA from X. laevis present in its urine could be transferred to Pr. xenopodis eggs…“ I think, the possibility cannot be excluded altogether, but the suggested process does not seem very likely. For one thing, the concentration of extracellular DNA in the urine would be very low, and the DNA could hardly penetrate the intact monogenean eggs, which are known to be impervious to medical treatment.

Answer: Please note that the penetration of DNA from anuran urine into the worm egg was just one of the possibilities of transmission we analyzed in our text. The first via of DNA transmission we explored in the main text aligns to the “you are what you eat” hypothesis from Doolittle (1998), which has been commonly evoked to explain HGT between host and parasites/blood feeding species (Gilbert et al. 2010, 2012; Wijayawardena et al. 2013 and references therein; Sibbald et al. 2020 and references therein). Given that we cannot exclude the possibility of DNA transmission to the egg (as also scored by the Reviewer), we chose to consider also this hypothesis in our text. We made changes in this paragraph to make these points clearer (lines 429 – 457).

Comment #4: Secondly, if the DNA did penetrate, how could it incorporate into the genome? As far as I know, in the well proven cases of incorporation (both in the nature and in the laboratory), the donor’s nucleic acid should be equipped with special sequencies. But the authors of the MS. either did not seek for or sought but did not find any symptoms of transposition. Could a non-specified recombination be the answer? We have to observe that the extraordinary similarity between X. laevis and Pr. xenopodis referred to short sequences containing 120 bp gene and only a part of the spacer, but no whole clasters. One would expect longer sequences for recombination. In page 14, lines 481-482, the authors write: “In addition, regarding the 5S rDNA evolution of anurans, recombination with the satellite DNA PcP190 should be considered.“ But, as I understand, that recombination with only 190 bp long repeat is also just hypothetical. At any rate, the authors of the MS mentioned no symptoms of such recombination (e.g. presence of the sequences from “hypervariable region“ of PcP190) or any other processes ensuring the integration of the frog DNA into Pr. xenopodis genome.

Answer: Indeed, we do not elucidate the molecular mechanism by which the hypothetical DNA from X. laevis was incorporated into the Pr. xenopodis genome. However, this kind of explanation has not been provided in most reported cases of HGT involving eukaryotes, including those largely accepted as good examples of HGT. In a recent review, Keeling noted: “Another major gap in our knowledge is mechanism: we conclude HGT happened, and sometimes plausibly explain why, but we almost never know how it happened” (Nature Reviews Genetics, 25: 416–430, 2024). We rephased lines 431-444 of the MS to better contextualize it.

Certainly, HGT mediated by transposable elements (TEs) is a well-known possibility, but it is not the only one (please see the next paragraph), and not identifying a TE responsible for the transfer is not sufficient to discard the hypothesis of a transfer. In our analysis, we searched for TEs in the Pr. xenopodis genome assembly, which is highly fragmented (N50 = 2,899; total number of scaffolds = 316,398), and we did not find any sequences that would typically facilitate the incorporation of foreign DNA, such as TIRs and/or LTRs of classical TEs. Additionally, we searched for the presence of TEs from Xenopus (including highly copy number TEs) in the genome of Pr. xenopodis, as these might be expected in the alternative scenario of DNA contamination and found no evidence for this (as pointed out in section 3.3).

Illegitimate recombination between rDNA sequences could certainly explain the insertion of a frog DNA into the flatworm genome. According to Rubinitz and Subramani (1984), high recombination frequencies may be expected between segments of 200 bp or longer, but recombination events can still occur between sequences shorter than 200 bp (Ayares et al. 1986; Rubinitz and Subramani 1984). Therefore, the 5S rRNA gene segment, which is composed of 120 bp, would be sufficient to enable recombination between tandemly repeated 5S rDNA units. Recently, using PCR with primers specific to 5S rDNA, we isolated from the genome of a species of the frog genus Lysapsus a chimeric fragment formed by juxtaposed 5S rDNA and satDNA PcP190 sequences (Targueta et al., 2023). Additionally, using PCR, we also isolated from the genomes of Lysapsus and Pseudis species fragments containing different types of PcP190 sequences, whose similarity was limited to a 120 bp region (Gatto et al., 2018). These findings suggest the occurrence of occasional recombination events both between subfamilies of the satDNA PcP190 and between this satDNA and 5S rDNA. Therefore, these findings are consistent with the hypothesis of recombination between 5S rDNA from Xenopus and Pr. xenopodis, since these 5S rDNAs likely share enough similarity to enable recombination (similar to what has been observed between 5S rDNA and PcP190 satDNA). In the MS, we cite these findings in the section we discuss about the forces involved in 5S rDNA evolution just as a reminder of illegitimate recombination affecting 5S rDNA families. However, recombination between 5S rDNA and PcP190 satDNA specifically cannot be considered as a mechanism responsible for incorporating 5S rDNA from Xenopus into the flatworm genome, as this satDNA has only been found in the anuran superfamily Hyloidea (being absent from Xenopus).

Other relevant points we would like to highlight here are: 1) extrachromosomal rDNAs are largely known, being found in several organisms (Cohen & Segal, 2009; Bughio & Maggert, 2019), including Xenopus (Reeves, 1978); 2) recombination between rDNA sequences are largely expected, being one of the factors to drive concerted evolution of this type of sequences  (Cohen et al., 2010; Eickbush & Eickbush, 2007); 3) recombination between extrachromosomal rDNA and genomic DNA is largely accepted, being commonly evoked to explain rDNA amplification in nuclear genomes (Cohen et al., 2010; Yüksel & Altungöz, 2023).  

Taken together, these arguments may support the idea that illegitimate recombination should be considered a possible mechanism to explain the hypothetical incorporation of Xenopus 5S rDNA into the Pr. xenopodis genome.

Comment #5: Thirdly, if the exogeneous sequences could find their way into the worm’s genome, why should they be settled there and not be blocked or deleted? Why should they propagate in the population (and eventually in the entire species) to the exclusion of the host’s variants? Do they confer the selective value comparable to that of the drug resistence plasmids?

Answer: The 5S rDNA originally from the worm (i.e., the host variant) was not excluded from the genome. As pointed out in lines 171-175 and shown in Table 1, we identified four 5S rDNA sequences in the Pr. xenopodis genome assembly and three of them clustered together with 5S rDNA sequences from other worms (as shown in Figure 1 of our MS).

The assembly of repetitive DNA regions has been one of the major technical problems in generating long arrays (reviewed in Treangen and Salzberg 2012). Because of this, we cannot provide a proper comparative analysis of the number/amount of 5S rDNA sequences present in this genome. Nevertheless, the maintenance (or even amplification/propagation) of 5S rRNA genes originally from Xenopus in the worm genome would not pose a problem, since the transcribing region of Xenopus 5S rDNA is ~ 70% similar to that of flatworms (as shown in Table 1 of our MS). In other words, the maintenance – or even the amplification of this gene – could have occurred through genetic drift (i.e., independently of selection).  

Comment #6: Fourthly, HGT in the repetitive genes, such as 5S rDNA, would be still more problematic, since the new variant should supplant not just one unit, but all the clusters, perhaps situated on different chromosomes. Here I would observe that the MS. has to provide an additional information about the organization of 5S rDNA in the studied parasitic worms. A simple figure would do showing the number of copies, the number of clusters, the number of chromosomes carrying these clusters, the situation of 5S rDNA in relation to other ribosomal genes (for they may be located on different chromosomes or in the same locus). All this data are relevant to the case.

Answer: Our original MS already provided the number of 5S rRNA gene copies we found in the Pr. xenopodis genome assembly (i.e., four copies), as well as a comparison among them (please see lines 171-175, Table 1, Figure 1). Each of the copies was found on a separate scaffold/contig (i.e., scaffolds/contigs 0184163, 0082537, 0013821, and 00070052, as presented in Table 1). The identification of all the scaffolds/contigs that had 5S rDNA is provided in Table 1. In the original MS, this information appeared at the bottom of the table, but in the revised MS we moved it to the end of the legend to improve visibility. Unfortunately, since the available Pr. xenopodis genome is not a chromosome-level assembly, it is not possible to determine which chromosomes harbor the 5S rDNA sites.

It is also important to note that, in the Pr. xenopodis genome assembly, we did not find only the 5S rDNA that was similar to the Xenopus gene; we also found 5S rRNA genes (in three scaffolds/contigs) that were very similar to the 5S rRNA genes of other worms (and clustered with them in Figure 1), as indicated in lines 171-175, in Table 1, and in Figure 1. Therefore, since the hypothetical scenario considered by the Reviewer – in which the Xenopus gene would have supplanted the original worm genes – was not supported by our findings (as we found 3 sequences that were clustered with those from other worms, as shown in Figure 1), we respectfully disagree that this should be considered a problematic issue.  

Comment #7: Finally, no mention is made in the MS. of what might be a more plausible alternative to the HGT hypothesis, namely, similarity of sequence determined by the similarity of function. Thus animals employing echolocation have not only analogous physiolological features but also remarkable similarity of the DNA sequences engaged in the process. Should we discuss a gene transfer between bats and dolphins? Or should we better look for the analogous functions (perhaps not ribosomal) of the similar NTS sequences?

Answer: Based on the teasing question presented here, it seems we probably didn’t make clear to the reader some important points regarding 5S rDNA structure and evolution.  Although a regulatory role in gene transcription has been attributed to the NTS of 5S rDNA, some studies have evidenced that only small portions of the NTSs effectively play some regulatory role (Korn & Brown, 1978; Nederby-Nielsen et al., 1993; Hallenberg & Frederiksen, 2001), as the major transcription control region (known as ICR – internal control region) is located inside the transcribing region (Pieler et al., 1987). Therefore, NTS is expected to have low adaptive value, which agrees with empirical data, since NTS is shown to be highly variable in both size and nucleotide sequence, even within the same genome (for some references, please see Merlo et al., 2013; Vierna et al., 2013; Targueta et al., 2023). In our analysis, we found extensive similarity between the NTS of 5S sequences found in Xenopus and Pr. xenopodis (lines 236-239; Figure 3), which is precisely what supports the HGT hypothesis over the convergence one.

We included a paragraph in the revised MS to assess this issue (lines 416-428).

Comment #8: Page 9, line 257-258. “…rDNA cloned sequences (X05264 and X02995, the latter of which contains the transcribed region of the precursor rRNA).“

X02995 is missing in Table S4, which shows alignment similarity”

Answer: The information regarding the sequence X02995 was included in Table S4.

Comment #9: Page 9, line 259. “…the 5’-end of the second enhancer…“

It is actually 3‘ end.

Answer: We corrected it in the revised MS.

Comment #10: Page 9, line 291. “A third region with a significant amount of mapped reads.”

It is not quite clear what are the first two.

Answer: We rephased some sentences and moved this paragraph to the previous one to make this clearer.

Comment #11: Fig. 5. What are the numbers at the ordinate?

Answer: Numbers at the ordinate indicate read mapping densities. We added this information in the figure legend.

Comment #12: Discussion. For the convenience of the reader, it would be advisable to supply references to the figures and tables throughout the Discussion.

Answer: Done.

Comment #13: Page 13, lines 419-420. “In conclusion, we cannot find any evidence refuting the HGT hypothesis to explain the discovery of a 5S rRNA gene identical to that of X. laevis in the genome of Pr. xenopodis

Two different ideas seem to be confused in this sentence: plausibility of HGT hypothesis and finding an alternative hypothesis to explain the data.

Answer: We rephrased this sentence in the revised MS.

Comment #14: Page 13, line 421. “According to this scenario,…“

It is not clear what this phrase refers to, for the word “scenario“ was mentioned previously in reference to the contamination.

Answer: We changed the main text in this section in the new version of the manuscript.

Comment #15: To conclude, if the authors insist on the claim in the title of the MS., I would suggest a major revision that would provide convincing facts in support of the HGT hypothesis. Otherwise, I think the authors could just change the title (e.g. by using question mark at the end would), expound the above-mentioned difficulties attending the HGT hypothesis, and discuss the alternative interpretation of the data. The MS. thus would be no less interesting, but more correct.

Answer: We changed the title and several sentences of the main text based on the reviewer’s comments.

References

Ayares D, Chekuri L, Song K-Y, Kucherlapati R (1986). Sequence homology requirements for intermolecular recombination in mammalian cells. Proc Natl Acad Sci USA, 83: 5199-5203.

Bughio F, Maggert KA (2019). The peculiar genetics of the ribosomal DNA blurs the boundaries of transgenerational epigenetic inheritance. Chromosome Res, 27: 19–30.

Cohen S, Segal D (2009). Extrachromosomal circular DNA in eukaryotes: possible involvement in the plasticity of tandem repeats. Cytogenet Genome Res, 124: 327–338.

Cohen S, Agmon N, Sobol O. et al. (2010). Extrachromosomal circles of satellite repeats and 5S ribosomal DNA in human cells. Mobile DNA, 1: 11.

Doolittle WF (1998). You are what you eat: a gene transfer ratchet could account for bacterial genes in eukaryotic nuclear genomes. Trends Genet, 14: 307-311.

Eickbush TH, Eickbush DG (2007). Finely orchestrated movements: evolution of the ribosomal RNA genes. Genetics, 175: 477–485.

Gatto KP, Mattos JV, Seger KR, Lourenço LB (2018). Sex chromosome differentiation in the frog genus Pseudis involves satellite DNA and chromosome rearrangements. Front Genet, 9: 301.

Gilbert C, Schaack S, Pace-II JK, Brindley PJ, Feschotte C (2010). A role for host-parasite interactions in the horizontal transfer of transposons across phyla. Nature, 464: 1347-1350.

Gilbert C, Hernandez SS, Flores-Benabib J, Smith EN, Feschotte C (2012). Rampant horizontal transfer of SPIN transposons in squamate reptiles. Mol Biol Evol, 29: 503-515.

Hallenberg C, Frederiksen S (2001). Effect of mutations in the upstream

promoter on the transcription of human 5S rRNA genes. Biochim Biophys Acta, 1520: 169-173.

Keeling PJ (2024). Horizontal gene transfer in eukaryotes: aligning theory with data. Nature Rev Genet, 25: 416-430.

Korn LJ, Brown DD (1978). Nucleotide sequence of Xenopus borealis oocyte 5S

DNA: Comparison of sequences that flank several related eukaryotic genes. Cell, 15: 1145-1156.

Merlo MA, Cross I, Manchado M, Cárdenas S, Rebordinos L (2013). The 5S RDNA High Dynamism in Diplodus Sargus Is a Transposon-Mediated Mechanism. Comparison with Other Multigene Families and Sparidae Species. J. Mol. Evol. 2013, 76: 83–97.

Nederby-Nielsen J, Hallenberg C, Frederiksen S, Sorensen PD, Lomholt B. (1993). Transcription of human 5S rRNA genes is influenced by an upstream DNA sequence. Nucleic Acids Res, 21: 3631–3636.

Pieler T, Hamm J, Roeder RG (1987). The 5S gene internal control region is composed of three distinct sequence elements, organized as two functional domains with variable spacing. Cell, 48: 91-100.

Reeves R (1978). Nucleosome structure of Xenopus oocyte amplified ribosomal genes. Biochemistry, 17: 4908-4916.

Rubinitz J and Subramani S (1984). The minimum amount of homology required for homologous recombination in mammalian cells. Mol Cell Biol, 4: 2253-2258.

Sibbald SJ, Eme L, Archibald JM, Roger AJ (2020). Lateral gene transfer mechanisms and Pan-genomes in eukaryotes. Trends in Parasitology, 36: 927-941.

Targueta CP, Gatto KP, Vittorazzi SE, Recco-Pimentel SM, Lourenço LB (2023). High diversity of 5S ribosomal DNA and evidence of recombination with the satellite DNA PcP190. Gene, 851: 147015. doi: doi.org/10.1016/j.gene.2022.147015.

Treangen TJ and Salzberg SL (2012). Repetitive DNA and next-generation sequencing: computational challenges and solutions. Nature Rev Genet, 13: 36-46.

Vierna J, Wehner S, Höner zu Siederdissen C, Martínez-Lage A, Marz M (2013). Systematic analysis and evolution of 5S ribosomal DNA in metazoans. Heredity, 111: 410-421.

Wijayawardena BK, Minchella DJ, DeWoody JA (2013). Host, parasites, and horizontal gene transfer. Trends in Parasitology, 29: 329-338.

Yüksel A, Altungöz O (2023). Gene amplifications and extrachromosomal circular DNAs: function and biogenesis. Mol Biol Rep, 50: 7693–7703.

Reviewer 2 Report

Comments and Suggestions for Authors

In silico analysis based on GenBank sequences obtained by different authors for undocumented DNA samples should not be considered as definitive evidence for the presence of HGT between frogs and flatworm. In the article much attention was devoted to examining evidence that rejects the assumption of contamination of Protopolystoma with Xenopus DNA. (A similar analysis has not been performed for the putative 5S rDNA from Pseudis to nematodes.) However, to further confirm the existence of HGT, taking into account the intragenomic heterogeneity of 5S rRNA genes, targeted sequencing of the these genes should be carried out from Xenopus and thoroughly washed Protopolystoma samples, possibly with DNase. Higdhly variable ITS of rDNA may serve as a marker for possible contamination of Protopolystoma sequences with Xenpus DNA.

It is unclear whether the suspected HGT case is isolated, transitory, or transmitted to offspring.

Thus, the existence of HGT between amphibians and worms remains a hypothesis for now. Therefore, it would be better to title the article “Data on HGT between…”, and add the corresponding phrases to the Abstract and Conclusions.

Supplementary figures and tables were not submitted for review, making it impossible to fully evaluate the MS. The most important Supplementary data should be presented in the main text of the article.

Author Response

Response to reviewer #2

We thank the reviewer for the analysis of our MS and for the helpful suggestions. We modified the main text based on the review and answer below each comment of the reviewer.

Comment #1: In silico analysis based on GenBank sequences obtained by different authors for undocumented DNA samples should not be considered as definitive evidence for the presence of HGT between frogs and flatworm. In the article much attention was devoted to examining evidence that rejects the assumption of contamination of Protopolystoma with Xenopus DNA. (A similar analysis has not been performed for the putative 5S rDNA from Pseudis to nematodes.) However, to further confirm the existence of HGT, taking into account the intragenomic heterogeneity of 5S rRNA genes, targeted sequencing of the these genes should be carried out from Xenopus and thoroughly washed Protopolystoma samples, possibly with DNase. Higdhly variable ITS of rDNA may serve as a marker for possible contamination of Protopolystoma sequences with Xenpus DNA.

Answer: We totally agree that in silico analysis does not provide definitive evidence for the presence of HGT. As we could not obtain isolated Protopolystoma samples to make further investigation, in the original MS we had tried to analyze the alternative hypothesis of contamination, aiming to provide some arguments that could support the HGT hypothesis. In order to avoid any misinterpretation, we have changed several sentences across the main text, including the title, to make clearer that we propose that HGT is the most likely scenario to explain the presence of Xenopus 5S rDNA among Protopolystoma xenopodis genome assembly, although definite evidence is still needed. Regarding the attention devoted to DNA contamination of P. xenopodis genome assembly, it was possible because genomic data are available for this species and X. laevis. Since it was not possible to identify the possible source of nematode contamination for the case regarding the Pseudis 5S rDNA, a similar attention to this case is not currently possible.

We also agree that the variable NTSs are special markers for the analysis of possible contamination. That is the reason why we highlighted in our text that i) not only the gene sequence but also the NTS of the rDNA sequence of Pr. xenopodis was found in the rDNA of Xenopus species and ii) other variable repetitive sequences are not shared between Xenopus and Pr. xenopodis.

Comment #2: It is unclear whether the suspected HGT case is isolated, transitory, or transmitted to offspring.

Answer: Pr. xenopodis genome assembly was made from sequencing libraries derived from different individuals and tissues (eggs, larvae, and adults). Since we mapped reads from both libraries used to assemble the Pr. xenopodis genome to a chromosome region annotated for the somatic 5S rDNA of X. laevis, it was expected that the frog gene that was suspectedly incorporated by the flatworm was transmitted to the offspring. In the revised version of the MS we made clear in the material and methods that the sequencing libraries used for this analysis are derived from different individuals.

Comment #3: Thus, the existence of HGT between amphibians and worms remains a hypothesis for now. Therefore, it would be better to title the article “Data on HGT between…”, and add the corresponding phrases to the Abstract and Conclusions.

Answer: In the reviewed version of the MS, we have changed the title and we rephrased several sentences in the main text to make it clear.

Comment #4: Supplementary figures and tables were not submitted for review, making it impossible to fully evaluate the MS. The most important Supplementary data should be presented in the main text of the article.

Answer: We are very sorry for that. The Supplementary Figures and Tables are available in the submission portal as a pdf file.

Round 2

Reviewer 1 Report

Comments and Suggestions for Authors The MS. has been improved, but it still does not take seriously the alternative hypothesis. The authors dismissed it by using dubious arguments (lines 419-425): that functions have not been found; that NTS is highly variable; that NTS has a “low adaptive value.“ Firstly, if the function has not been found, it does not mean that it does not exist. In fact, the recent studies keep discovering new functions in what used to be termed “junk DNA“. For example, in the human nucleolar NTS, there are coding sequences for several micro RNAs and lncRNAs involved in various ribosomal and non-ribosomal functions, as well as multiple putatively functional units. Secondly, the mere quantitative level of variability in a locus does not determine its functionality. Thus the number of SNVs in 28S rDNA coding sequence is as high as in the most variable parts of NTS. Thirdly, whatever the adaptive value of NTS may be, this does not relate to the question directly. As the authors will know, whereas some functions are indispensable, others can be more or less easily replaced by alternatives. But the functions of low adaptive value play their role nonetheless. In short, the authors have either to accept the hypothesis as a plausible alternative or to refute it by convincing arguments. And yes, the authors do imply that HGT is a common phenomenon. Indeed, seventeen species (taken from the available databases) were examined and the HGT events were suspected in two of them. This is more than 10%. I think, the extraordinary inference will need some comment.

Author Response

Comment #1: The MS. has been improved, but it still does not take seriously the alternative hypothesis. The authors dismissed it by using dubious arguments (lines 419-425): that functions have not been found; that NTS is highly variable; that NTS has a “low adaptive value.“ Firstly, if the function has not been found, it does not mean that it does not exist. In fact, the recent studies keep discovering new functions in what used to be termed “junk DNA“. For example, in the human nucleolar NTS, there are coding sequences for several micro RNAs and lncRNAs involved in various ribosomal and non-ribosomal functions, as well as multiple putatively functional units.

Response: Although it is certainly possible that the NTSs harbor unknown and exciting functions, we cannot simply assume their existence. On the other hand, several comparative analyses of 5S rDNA sequences from diverse organisms have revealed conserved transcribing (or “coding”) regions contrasting with the variable intergenic non-transcribing spacers. The important point here is that the NTSs (which is expected to be much more variable than the transcribing region) found in Xenopus and its parasite were also very similar. Based on the available data (which includes no evidence for any functional role employed by the whole extension of the shared NTS sequence), we have no argument that supports the hypothesis of convergent evolution. We rephrased lines 426-428 to make it clearer.

Comment #2: Secondly, the mere quantitative level of variability in a locus does not determine its functionality.

Response: We completely agree. This aligns with our response to comment #1.

Comment #2: Thus the number of SNVs in 28S rDNA coding sequence is as high as in the most variable parts of NTS.

Response: Unfortunately, we are unable to fully understand this point, as we could not identify which NTSs the reviewer considered in this analysis. Numerous studies that compare 5S rDNA transcribing regions with their NTSs have shown that the NTSs are much more variable than the transcribing regions. While the 5S rDNA transcribing regions are highly conserved, several types of NTSs have been found even within the same genome (Vierna et al., Biochem. Genet. 47, 635–644, 2009; Fujiwara et al., Genetica 135, 355–365, 2009; Vozárová et al., Front. Plant Sci. 12, 343, 2021). Frequently, the different types of 5S rDNA NTSs are easily distinguished by their size. Lastly, it is worth noting that variation in 5S rDNA has important implications for interspecific analyses, as it has already been used for the taxonomic identification of species with forensic and commercial relevance (e.g., Bertea et al., Phytochemistry 67, 371–178, 2006; Merlo et al., Genes Genet. Syst. 85, 341-349, 2010).

Comment #3: Thirdly, whatever the adaptive value of NTS may be, this does not relate to the question directly. As the authors will know, whereas some functions are indispensable, others can be more or less easily replaced by alternatives. But the functions of low adaptive value play their role nonetheless. In short, the authors have either to accept the hypothesis as a plausible alternative or to refute it by convincing arguments.

Response: While we cannot properly assign any functional role to the NTSs, nor find any evidence of microRNA or lncRNA transcription from them, we also cannot assume such elements exist (nor they existed and were replaced). Therefore, attributing any hypothetical function to the NTS cannot be considered a convincing argument in favor of the hypothesis of convergent evolution. In the absence of evidence for any functional role in the NTSs, this cannot be used to explain the similarity observed between the NTS sequences found in anurans and their parasites.    

Comment #4: And yes, the authors do imply that HGT is a common phenomenon. Indeed, seventeen species (taken from the available databases) were examined and the HGT events were suspected in two of them. This is more than 10%. I think, the extraordinary inference will need some comment.

Response: Unfortunately, we cannot understand which are the 17 species considered in this comment. In any case, we would like to emphasize that our goal in this MS was to point out the intriguing similarity observed between the 5S rDNA sequences found in certain frogs and parasites. We examined in detail the case of Xenopus laevis and its parasite Xenopodius protopolystoma. Additionally, we indicated another case for further studies.  

We also highlight that we analyzed the 5S rDNA sequences from 36 anuran species (i.e., all currently available sequences in public databases), which represent 11 families, encompassing ditantly related clades.

Finally, we would like to emphasize the importance of reporting the observed similarity between DNA sequences attributed to anurans and parasites in the available databases, as this is not an expected finding. Given that contamination could plausibly explain such similarity, we believe that documenting this finding and analyzing its possible causes is an important contribution.